# RETRACTED: Post-Traumatic Growth and Quality of Life among World Trade Center Health Registry Enrollees 16 Years after 9/11

**DOI:** 10.3390/ijerph19159737

**Published:** 2022-08-08

**Authors:** Howard E. Alper, Leen Feliciano, Lucie Millien, Cristina Pollari, Sean Locke

**Affiliations:** 1World Trade Center Health Registry, New York City Department of Health and Mental Hygiene, Division of Epidemiology, New York, NY 11101, USA; millienl@finance.nyc.gov (L.M.); cdpollari@gmail.com (C.P.); seanhlocke@gmail.com (S.L.); 2Graduate School of Public Health and Health Policy, The City University of New York, New York, NY 10027, USA; leen.feliciano43@sphmail.cuny.edu

**Keywords:** 9/11, post-traumatic growth, World Trade Center, quality of life, SF-12, physical function, mental function

## Abstract

A recent study of World Trade Center Health Registry enrollees found that about one-third experienced post-traumatic growth (PTG) in the wake of the 9/11 attacks and that PTG was associated with social support and social integration. However, the implications of PTG for the enrollees’ overall quality of life are unknown. The present study investigated the prevalence of PTG and its association with the SF-12 physical and mental functioning quality of life scales in a sample of 4760 enrollees from the Registry’s Health and Quality of Life Study (HQoL) who completed the first four surveys, were older than 18 on 9/11, reported English as their primary spoken language, and provided consistent self-report of 9/11 physical injury at the Registry’s baseline and HQoL surveys. We employed multivariable linear regression to evaluate the association between PTG and the SF-12 physical and mental scales, controlling for sociodemographic and other variables. We found that 31% of the sample enrollees experienced PTG and that PTG exhibited a clinically and statistically significant association with the SF-12 mental scale but not the physical scale (physical: b = 0.15 (−0.45, 0.75), mental: b = 3.61 (2.85, 4.37)). Those who were physically injured during 9/11 showed larger improvements in mental functioning than those who were not. PTG has implications for the overall mental quality of life that should be further investigated.

## 1. Introduction

The World Trade Center (WTC) terrorist attacks on 11 September 2001 resulted in over 2700 deaths and many thousands more physically injured. The dust/debris cloud generated by the collapse of the WTC towers enveloped many survivors and responders, who also may have been physically injured or witnessed horrific events such as seeing airplanes strike the towers. These exposures have been found to be associated with a variety of physical and mental conditions, such as asthma [1,2,3,4,5], post-traumatic stress disorder (PTSD) [6,7,8,9], heart disease [6,10], stroke [8], cancer [11,12], autoimmune disease [13], and somatic symptoms [14].

Research by the Registry and others has shown that exposure to traumatic events can have life-long implications such as PTSD, but recent studies in the literature have discovered variations in how individuals respond to trauma [15,16]. Some individuals report marked improvements in their quality of life (physical and mental functioning) following a traumatic event [17,18,19]. The positive psychological changes that occur in response to the traumatic event may result in a higher level of functioning, meaning that the individuals do not return to the baseline or pre-trauma levels of functioning, as would be the case with resilience, but instead surpass pre-trauma functioning [20,21].

This phenomenon, known as post-traumatic growth (PTG), was originally developed by Tedeschi and Calhoun and has gained widespread attention in public health research [22,23]. It is also sometimes referred to as “benefit finding”, “growth experience”, “adaptive growth”, “stress-related growth”, or “positive psychological changes” [22,23,24,25]. PTG can include a positive shift in mindset and beliefs, greater life satisfaction, acceptance of the traumatic event, and improved interpersonal relationships. Individuals who report PTG may also engage in new or changed spiritual beliefs and assign positive meaning to their traumatic experiences. Tedeschi and Calhoun developed the Post-traumatic Growth Inventory (PTGI), a 21-item scale to assess five domains of PTG—new possibilities, relating to others, personal strength, spiritual change, and appreciation of life [22].

The extent to which individuals experience PTG is determined by the level of adaptive response to the traumatic event [23]. In other words, PTG is not the direct result of trauma but rather a result of the psychological process and struggle to accept a new reality and positively interpret traumatic events. Additionally, the severity of trauma determines the likelihood of PTG. Mild or severe trauma tends to result in adverse health outcomes or PTSD, whereas moderate trauma can result in PTG if individuals cope positively. PTG may be time-sensitive, emerging shortly after a traumatic event and increasing with time [26]. PTG has been observed in various populations following a traumatic event, including veterans [27], cancer survivors [26,28,29], college students experiencing trauma [30,31,32], sexual abuse survivors [33], and terror attack survivors [34,35,36]. 

Recent research has shown that 34.3% of World Trade Center Health Registry (WTCHR) enrollees demonstrated moderate-to-high PTG in the aftermath of direct exposure to the 9/11 attacks [37]. Previous studies on PTG and 9/11-related exposures focused on indirect 9/11 exposure through television [35,36] or assessed personality and mental health factors associated with PTG in a 9/11 population with indirect 9/11 exposures [20]. However, the implications of PTG in the aftermath of direct disaster exposure for physical and mental health are unknown. Exploring the association between PTG and quality of life is useful to understand the long-term physical and mental health effects of 9/11 and to inform new policies and treatments. Further, it is important to investigate this association separately for those with and without physical injuries on 9/11, because injury on 9/11 has been shown to be associated with the subsequent development of PTSD [38] and heart attack. Diseases such as these could affect both PTG and quality of life and, therefore, their association.

The mechanism underlying the relationship between PTG and health-related quality of life (HRQOL) is not well-understood. HRQOL is a concept that refers to one’s perceived physical and mental health status. Several systematic reviews have evaluated PTG and HRQOL but have found conflicting results. One review did not find a statistically significant association between PTG and the physical functioning domain of HRQOL [39]. Another review found that half the studies included revealed no relationship between PTG and HRQOL, while others found a positive relationship [40]. A recent review from 2020 found a positive relationship between PTG and HRQOL among cancer patients, indicating that PTG may play an integral part in successful coping [41]. More research is needed to gain insight into the complex relationship between PTG and HRQOL. The aims of this study were to (1) determine the prevalence of post-traumatic growth (PTG) among World Trade Center Health Registry (WTCHR) enrollees with direct exposure to 9/11, (2) evaluate the association between PTG and physical and mental functioning in a sample of physically injured plus non-injured enrollees, and (3) evaluate this association separately for physically injured and non-injured enrollees. 

## 2. Materials and Methods

### 2.1. Participants

The New York City Department of Health and Mental Hygiene (NYC DOHMH) established the WTCHR in 2002 to monitor the long-term physical and mental health effects of the attacks on 11 September 2001 (9/11). The Registry conducted its initial survey in 2003–2004 (Wave 1), which included rescue and recovery workers, area residents, area workers, passersby, and students and staff in local schools. An additional three health surveys have been conducted: Wave 2 in 2006–2007, Wave 3 in 2011–2012, and Wave 4 in 2015–2016. Details describing the Registry’s methods for recruitment and data collection are available in previous publications [3,42]. 

In 2017–2018, the Registry conducted the Health and Quality of Life (HQoL) survey, to investigate the potential association between physical injury on 9/11 and quality of life, using measures for the latter such as the SF-12 overall physical and mental functioning [21], the SSS-8 somatic symptoms scale, the GRIT scale, Post-Traumatic Growth Inventory [22,23], etc. Participation was limited to enrollees who had completed all four Registry survey waves fielded to date, were at least 18 years of age at Wave 1, and whose primary spoken language was English. Two groups were included in the survey. The first group consisted of all enrollees who reported having one or more of the following 9/11-related physical injuries on the Wave 1 survey: cut, abrasion, or puncture wound; sprain or strain; burn; broken bone or dislocation; and concussion or head injury. Those who reported “other injury” or “eye injury” were excluded. The second group consisted of a random selection of enrollees from the Wave 1 survey who reported no physical injury during Wave 1. This second group was chosen to be similar in size to the first, to keep the total HQoL sample relatively small compared with the full Registry. Furthermore, for the present study, only enrollees who reported consistent responses for sustaining a 9/11-related physical injury on both Wave 1 and the injury and HQoL survey were included in the current analysis (e.g., “yes” on Wave 1 and “yes” on the injury and HQoL survey or “no” on both). Those who did not have complete data on the exposures, outcome, and covariates were excluded. No significant differences were observed between those who were included and excluded from the study. The Institutional Review Boards at the Centers for Disease Control and Prevention and the NYC DOHMH approved the Registry’s protocol and use of data. Further details regarding the creation of the HQoL survey including recruitment and outcomes are available in a prior Registry publication [43]. 

### 2.2. Outcome

The HQoL survey included a series of 12 questions known as the Short-Form Health Survey-12, Version 1 (SF-12) [21]. The SF-12 was derived from the SF-36 to provide an efficient method for assessing overall physical and mental health functioning through a mean physical health component summary score (PCS-12) and a mean mental health component summary score (MCS-12). The summary scores are based on combinations of SF-12 questions that were identified as representing the overall physical and mental status and that were highly correlated with the concomitant SF-36 scores. For both the physical and mental summary scores, higher values correspond to better, healthier functioning.

### 2.3. Exposure

Post-traumatic growth was evaluated using the Post-Traumatic Growth Inventory (PTGI), a self-report 21-item scale developed by Tedeschi and Calhoun to assess five domains of positive change following trauma: new possibilities, relating to others, personal strength, spiritual change, and appreciation of life [22,23]. Participants specified their level of agreement with each item on the PTGI, which involved statements about positive change experienced after trauma, such as “I have a greater appreciation for the value of my own life” and “I have more compassion for others”. PTGI Item 7, “I established a new path for my life”, was unintentionally omitted from the HQoL survey, so it was not distributed to participants. As a result, PTGI Item 7 was excluded from all analyses. A total PTGI score was calculated from the sum of the remaining 20 PTGI items. The PTGI uses a 5-point Likert scale to rank responses from 0 (“I did not experience this change as a result of my crisis”) to 5 (“I experienced this change to a very great degree as a result of my crisis”). Therefore, the total PTGI score ranged from 0 to 100. Average PTGI scores (total/20) ≥ 3 were taken as indicative of moderate-to-high post-traumatic growth. 

### 2.4. Covariates

The demographic characteristics included in the current analysis include sex (male, female), age at 9/11, race (White Non-Hispanic, Black Non-Hispanic, Hispanic, Asian, or Multiracial/other), education (less than high school, high school/GED, some college, or college/post-grad), and income (0 ≤ USD 25 K, USD 25 ≤ USD 50 K, USD 50 ≤ USD 75 K, USD 75 ≤ USD 150 K, ≥ USD 150 K). 

Risk factors included employment status, social support, life-threatening events post-9/11, mental health conditions post-9/11, physical health conditions post-9/11, and alcohol or drug use post-9/11. Employment status was defined as “yes” if participants were employed full-time/part-time/self-employed during Wave 4 and as “no” if the enrollee reported otherwise. Information on social support was collected during Wave 3. Research has shown that little to no social support results in decreased mental functioning and worsens the severity of pre-existing mental health conditions such as PTSD, depression, and possibly somatic symptom disorder [14]. Participants were asked the following five questions to measure social support: “How often is someone available to take you to the doctor if you need to go?”, “to have a good time with you?”, “to hug you?”, “to prepare your meals if you are unable to do it yourself?”, and “to understand your problems?”. Responses were rated from 0 (“none of the time”) to 4 (“all of the time”). A total score for social support consisted of the sum of all five questions and ranged from 0 to 20, with low social support defined as a total score below 15. Similarly, exposure to life-threatening events can result in decreased mental functioning and add to mental health issues such as PTSD and depression [44]. Participants either confirmed or denied up to eight life-threatening events post-9/11 (such as having experienced a disaster, accident, or threat with a weapon) during Wave 3. The number of life-threatening events experienced was based on the total number of affirmative responses and categorically expressed as low (0 events), medium (1 event), high (2 events), and very high (3 to 8 events). Mental health post-9/11 was evaluated by asking participants if they experienced PTSD, anxiety, or depression between the baseline and Wave 4 surveys. Physical health post-9/11 was categorically expressed as having none, one, or two or more of the following chronic diseases: asthma, hypertension, angina, heart attack, stroke, diabetes, coronary heart disease, gastroesophageal reflux disease, rheumatoid arthritis, multiple sclerosis, or peripheral neuropathy, between the baseline and Wave 4 surveys. Research has shown that alcohol and drug misuse lead to poor physical and mental health status and may worsen pre-existing PTSD and depression [45,46,47,48]. Participants were asked the following question during Wave 4 to determine the problem of alcohol or drug use post-9/11: “Have you ever stayed overnight or longer at a hospital, rehabilitation facility, or mental health center so you could receive treatment or counseling for alcohol or drug use?” Participants either confirmed or denied a problem with alcohol and drug misuse and specified if this occurred before 9/11, after 9/11, or both before and after 9/11. Those who only reported alcohol and drug use before 9/11 were excluded from the current analysis.

### 2.5. Statistical Analysis

The primary outcomes of this analysis were the physical and mental health summary component scores (PCS-12 and MCS-12) of the SF-12. The mean PCS-12 and MCS-12 scores were evaluated for PTG, plus sociodemographic variables and risk factors, for the entire analytic sample and separately by physical injury status. The distributions for the above categorical variables were also calculated.

A bivariate analysis was conducted for the entire analytic sample to evaluate the association between PTG and physical and mental functioning.

The associations between PTG and overall physical and mental functioning were obtained from multivariable linear regressions of PCS-12 and MCS-12 on PTG, controlling for the above sociodemographic and risk factors. The resulting regression coefficients represented the extent of change in the PCS-12 or MCS-12 score relative to the reference category for the variable of interest. *p*-values and 95% confidence intervals for the beta were also calculated. The significance level was set at a 2-sided value of alpha < 0.05. All analyses were conducted using SAS 9.4 (SAS Institute, Inc., Cary, NC, USA).

## 3. Results

### 3.1. Demographic Characteristics of the Total Sample

The study sample consisted of 4760 enrollees. Most participants were White Non-Hispanic (78%), male (63%), and aged 25–44 at 9/11 (51%). Approximately 60% of participants completed a bachelor’s degree or post-graduate education, and 64% reported employment during Wave 4 (2015–2016), with an average annual income greater than USD 75,000 USD but less than USD 150,000 USD during Wave 1 (Table 1).

More than half of the participants reported moderate-to-high levels of social support (58%) and a low number of threatening events (66%) at Wave 3. Most participants reported no post-9/11 mental disease (71%) and no post-9/11 problem with alcohol or drugs (97%), and a minority reported no post-9/11 physical disease (44%) (Table 1).

### 3.2. Prevalence of PTG

The overall prevalence of PTG among WTCHR Enrollees in the sample was 31% (Table 1).

### 3.3. The Bivariate Associations of PTG with SF12

Participants with moderate-to-high PTG were more likely to show improved mental functioning in the bivariate analysis (β = 3.73, 95% CI: 2.96, 4.50; *p* ≤ 0.05) (Table 2). There was a small difference in physical functioning (β = −0.61, 95% CI: −1.20, −0.02; *p* ≤ 0.05).

### 3.4. Multivariate Associations of PTG with SF12

Previous research on the minimal clinically important difference between PCS-12 and MCS-12 in patients [49] revealed β ≥ 3.29 for both physical and mental functioning as clinically meaningful. Participants with moderate-to-high PTG were more likely to show clinically significant mental functioning in the multivariate analysis (β = 3.61, 95% CI: 2.85, 4.37; *p* ≤ 0.05). The association between PTG and mental functioning was enhanced in those participants who also experienced one or more physical injuries during 9/11 (β = 4.58, 95% CI: 2.98, 6.19; *p* ≤ 0.05). There was no association between PTG and physical functioning for the sample, the physically injured group, or the physically non-injured group (Table 2).

## 4. Discussion

This study evaluated the prevalence of post-traumatic growth and its association with overall physical and mental functioning. We found that 31% of the sample demonstrated PTG. In addition, PTG was found to be clinically and substantially related to mental functioning but not to physical functioning. The positive association observed between PTG and mental functioning supports the notion that PTG is a psychological process born out of a careful examination and positive interpretation of traumatic events [23,50,51]. This is also consistent with other studies that show an association between PTG and improved mental functioning in military personnel [52,53,54].

In terms of gender, other studies have reported lower MCS-12 and PCS-12 scores for women when measuring health-related quality of life using the SF-12 [55,56]. The differences observed for age are consistent with the literature on the SF-12 showing that mental health tends to improve with age, while physical health declines [57,58,59]. The role of education in promoting physical health and functioning has been well-documented in the literature [60,61,62]. The increased physical and mental functioning observed in conjunction with increases in annual income are to be expected given that income improves access to health care [63]. Our findings on social support and quality of life are consistent with the findings of other studies that link higher levels of social support to better physical and mental health outcomes [64,65,66]. Our finding that physical and mental health decline with increased traumatic events or threats is consistent with that reported in the literature [67,68,69]. The enhanced associations of PTG with both physical and mental functioning for physically injured enrollees may result from the increased adversity arising from physical injury, compared with non-injury.

Our results for the association of PTG with the SF-12 are consistent with the results of other investigations into the association of PTG with quality of life (QoL). For example, a study of North Korean defectors to South Korea [70] found that PTG was positively associated with a World Health Organization (WHO) QoL measure for quality of life. Another study, of the 2011 Tohoku earthquake in Japan [71], performed trajectory analyses of PTG and found three trajectories: no growth, illusory growth, and growth. QoL was seen to be positively associated with the probability of being in this sequence of PTG trajectories. Finally, another study of traumatized psychiatric patients [18] found that PTG was associated with QoL and that PTG explained more of the variance in the QoL outcomes than PTSD or any other variable in the model.

PTG has received much attention in disaster and other types of studies in the literature in recent years. However, doubts persist as to whether self-reported PTG represents “true” post-traumatic growth. One way to attempt to answer this question is to determine if self-reported PTG predicts outcomes of interest (e.g., QoL, self-reported chronic disease, hospitalizations). This study provides provisional answers to the above concerns, in that PTG was found to be associated with the SF-12 mental health but not the physical health measure of overall functioning. Future research could investigate the PTG–SF-12 association longitudinally, as well as the association of PTG with self-report and hospitalization records for physical and mental health diseases.

### Strengths and Limitations

A strength of this study is the use of a prospective cohort, which allowed us to investigate the association between exposures to, and sequelae of exposure to, the 9/11 attacks and later emerging conditions. We also had a sufficiently large sample size to ensure adequate power to detect the PTG–SF-12 association. Further, the sample was composed of physically injured and non-injured enrollees, which allowed the separate determination of the PTG–SF-12 association for these groups. The present investigation also used a well-validated measure of physical and mental functioning.

One limitation is that though the Registry is a cohort, the HQoL study was cross-sectional, so the effect of PTG on changes in physical and mental functioning could not be evaluated. A second limitation is that this study included only enrollees whose primary spoken language was English, which introduced a bias in our ability to generalize our results to the entire Registry population. A third limitation was that we treated gender as binary, so we could not investigate the PTG–SF-12 association for enrollees who identified as non-binary. Another limitation was that data for PTG, SF-12, and covariates were self-reported. However, there has been good agreement between Registry findings based on self-report and those based on hospitalization data, at least for exposure–disease associations [72]. Additionally, the HQoL sample is a non-random sample of the Registry population, so inferences from this study may not be generalized to the latter. Finally, the PTGI is self-reported, and previous research has shown only modest agreement with either more objective behavior measures or reports from the friends of the subjects [73,74].

## 5. Conclusions

We investigated the association between post-traumatic growth and overall physical and mental functioning in a sample of World Trade Center Health Registry enrollees. We found that post-traumatic growth was associated with the overall mental functioning but not physical functioning, in WTCHR enrollees 16 years post-9/11. Specifically, moderate-to-high PTG was found to be clinically and substantially related to mental functioning. Participants who experienced physical injuries during 9/11 showed greater levels of moderate-to-high PTG and a stronger association of PTG with overall mental functioning.

## Figures and Tables

**Table 1 ijerph-19-09737-t001:** Exposure and sociodemographic frequency distributions and physical and mental SF-12 scores.

Variable	Full Sample	Injured Only	Non-Injured Only
N	%	PCS-12	MCS-12	N	%	PCS-12	MCS-12	N	%	PCS-12	MCS-12
Post-Traumatic Growth												
Yes	1398	30.60	41.24	50.51	394	34.81	38.18	46.75	1004	29.22	42.38	51.92
No	3170	69.40	41.92	46.96	738	65.19	36.90	41.61	2432	70.78	43.33	48.47
Sex												
Male	2905	62.55	41.46	49.11	753	65.08	37.09	44.93	2152	61.71	42.94	50.52
Female	1739	37.45	42.14	46.18	404	34.92	37.91	40.23	1335	38.29	43.26	47.75
Age at 9/11												
0–17	20	0.43	48.25	33.45	1	0.09	---	---	19	0.54	48.25	33.45
18–24	197	4.24	46.51	46.40	29	2.51	44.20	40.32	168	4.82	46.83	47.24
25–44	2380	51.25	42.36	46.80	633	54.71	37.89	42.93	1747	50.10	43.90	48.13
45–64	1982	42.68	40.55	49.79	482	41.66	36.41	44.30	1500	43.02	41.75	51.39
65 or older	65	1.40	33.25	52.38	12	1.04	24.31	43.65	53	1.52	34.96	54.06
Race/Ethnicity												
White non-Hispanic	3636	78.29	41.96	48.59	839	72.52	37.52	44.35	2797	80.21	43.25	49.81
Black non-Hispanic	333	7.17	41.17	46.54	119	10.29	37.73	37.51	214	6.14	42.71	50.60
Hispanic	388	8.35	40.03	44.57	134	11.58	36.21	41.41	254	7.28	41.79	46.03
Asian	160	3.45	41.60	47.30	31	2.68	36.48	44.53	129	3.70	42.73	47.91
Other	127	2.73	40.69	46.98	34	2.94	36.88	43.92	93	2.67	41.79	47.87
Education												
High School/GED	750	16.19	39.36	47.46	253	21.94	35.07	42.17	497	14.28	41.15	49.66
Some College	1108	23.92	39.93	47.08	355	30.79	36.75	42.96	753	21.64	41.40	48.99
College/Post-Grad	2775	59.90	43.05	48.58	545	47.27	38.70	44.24	2230	64.08	44.06	49.59
Income at Wave 1												
<USD 25 K	235	5.54	41.10	42.79	92	8.65	34.30	35.79	143	4.50	44.47	46.26
USD 25–<50 K	655	15.44	41.30	45.96	192	18.06	37.23	41.34	463	14.56	42.78	47.64
USD 50–<75 K	976	23.00	40.77	47.42	233	21.92	35.69	42.57	743	23.36	42.24	48.82
USD 75–<150 K	1787	42.12	41.67	49.17	436	41.02	37.48	45.25	1351	42.48	42.97	50.38
USD 150 K or more	590	13.91	43.90	49.44	110	10.35	41.83	45.35	480	15.09	44.38	50.39
Employed at Wave 4												
Yes	2972	64.00	43.54	48.43	593	51.25	40.21	44.67	2379	68.22	44.34	49.33
No	1672	36.00	38.24	47.32	564	48.75	34.03	41.96	1108	31.78	40.18	49.79
Social Support Wave 3												
Yes	2662	58.22	42.59	51.02	546	48.32	38.77	47.34	2116	61.48	43.57	51.95
No	1910	41.78	40.49	43.92	584	51.68	35.94	39.49	1326	38.52	42.28	45.66
Threatening Events, Wave 3												
Low	3048	65.99	42.67	49.14	598	52.27	38.64	44.39	2450	70.50	43.58	50.21
Medium	828	17.93	40.94	46.28	222	19.41	36.80	41.73	606	17.44	42.36	47.85
High	367	7.95	39.75	45.86	131	11.45	36.72	42.25	236	6.79	41.40	47.82
Very High	376	8.14	37.68	45.36	193	16.87	34.58	43.27	183	5.27	40.67	47.38
Post-9/11 Mental Condition												
Yes	1347	29.01	40.00	39.98	595	51.43	36.63	38.13	752	21.57	42.54	41.38
No	3297	70.99	42.37	51.18	562	48.57	38.11	48.90	2735	78.43	43.19	51.61
Post-9/11 Physical Condition												
0	2030	43.71	44.50	50.08	278	24.03	42.09	46.15	1752	50.24	44.85	50.65
1	1374	29.59	42.03	47.87	342	29.56	39.05	44.08	1032	29.60	42.95	49.04
2 or more	1240	26.70	36.83	44.91	537	46.41	34.00	41.67	703	20.16	38.84	47.23
Post-9/11 Alcohol/Drugs												
Yes	134	3.29	39.12	39.05	55	5.57	35.33	35.56	79	2.56	41.86	41.57
No	3940	96.71	41.79	48.32	932	94.43	37.47	43.86	3008	97.44	43.09	49.66

**Table 2 ijerph-19-09737-t002:** Association between post-traumatic growth and physical and mental functioning.

Sample	Physical Functioning	Mental Functioning
	β	95% CI	β	95% CI
Full (Injured + Non-Injured) ^a^	−0.61	(−1.20, −0.02)	3.73	(2.96, 4.50)
Full (Injured + Non-Injured) ^b^	0.15	(−0.45, 0.75)	3.61	(2.85, 4.37)
Injured Only ^b^	0.92	(−0.34, 2.17)	4.58	(2.98, 6.19)
Non-Injured Only ^b^	0.00	(−0.67, 0.68)	3.30	(2.44, 4.16)

^a^—Bivariate regression; ^b^—multivariate regression controlling for sex, age, race and ethnicity, education, income, employment, social support, threatening events, post-9/11 mental health, number of post-9/11 physical health conditions, and drug/alcohol misuse.

## Data Availability

The data presented in this study are available on request from the corresponding author. The data are not publicly available due to subject confidentiality.

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
