# Peer review of "Post-Traumatic Growth and Quality of Life among World Trade Center Health Registry Enrollees 16 Years after 9/11"

_ijerph, 2022, doi:10.3390/ijerph19159737_

Round 1

Reviewer 1 Report

This paper provides a clear and concise account of a study investigating PTG outcomes among those affected by the World Trade Centre terrorist attacks. The findings contribute new knowledge to the PTG evidence base by providing a more nuanced understanding of associations between PTG and mental and physical health outcomes. This new knowledge informs strengths based approaches to disaster recovery. I have only three minor suggestions for this paper:

1. When using the terms 'injury' or 'injured' in the abstract and concluding statements it would be helpful to preface them with physical - i.e. 'physically injured' or 'physical injury' to very clearly differentiate it from mental or moral injury.

2. To acknowledge the use of binary sex terms male/female as a limitation 

3. There appears to be a leftover reminder (refs) in the last line of the strengths and limitations section that needs to be removed.

Author Response

Response to Reviewer 1 for “Posttraumatic Growth and Quality of Life Among World Trade Center Health Registry Enrollees 16 Years After 9/11”

This paper provides a clear and concise account of a study investigating PTG outcomes among those affected by the World Trade Centre terrorist attacks. The findings contribute new knowledge to the PTG evidence base by providing a more nuanced understanding of associations between PTG and mental and physical health outcomes. This new knowledge informs strengths-based approaches to disaster recovery. I have only three minor suggestions for this paper:

Comments:

  1. When using the terms 'injury' or 'injured' in the abstract and concluding statements it would be helpful to preface them with physical - i.e. 'physically injured' or 'physical injury' to very clearly differentiate it from mental or moral injury.

Response: We have made this correction throughout the manuscript.

  1. To acknowledge the use of binary sex terms male/female as a limitation 

Response: We have added tis concern to the “Strengths and Limitations” section.

  1. There appears to be a leftover reminder (refs) in the last line of the strengths and limitations section that needs to be removed.

Response: We have done so.

Reviewer 2 Report

This study is interesting, and builds upon a previous study by Registry researchers.  In addition, the theme is very well suited to this special issue.   It is well written by experts who have clearly been working for a long time with the very rich data gathered from members of the WTC Registry.  What follows are some questions to be addressed and points that need to be clarified.  

Line 21: Reference is made to those who were injured on 9/11.  I found out later in the paper that this refers to physical injury, but I think that the word “injury” is used a number of times before this is clarified in the methods section.  This should be clarified, especially since the paper is looking at both physical and mental health quality of life.

Line 31: “These exposures led to the development of ….”  I am not sure if all of these studies show direct causality - perhaps use the language of “associated with” when the studies do not necessarily show direct causality.

Line 70: You mention here the reason that this is an important study - this needs to be discussed and elaborated on in the discussion section and noted in conclusion as well.  I am not sure I fully understand how exploring this relationship is important for understanding the long term health impacts and would like this expanded upon later in the paper. 

Line 75 - One of your goals is to evaluate the association between PTG and mental/physical functioning in a sample of injured and non - injured patients - why would you separate out the two?  I think this needs to be addressed in the introduction as well - the reason that it is believed that these might be different - a theoretical basis would be great, or something that distinguishes these groups in past research?  Lots of research looks at the difference between those caught in the dust cloud or not - why was that NOT chosen?  (Not that it needed to be - but I am using that as an example of why we need to understand the reason that you are looking at these two groups.

Minor points: some small typos, and check fonts in the tables.  

Materials and methods: Line 88 - Can you please clarify what the HQoL is?  Is this something that is comprised only of the SF-12, or is the sf-12 embedded in this?  Please describe what this is. And if this particular survey was created by the Registry, should it be cited?

Line 110 - Just want to be sure you meant to use reference 20 here rather than 21.

OUTCOME - can you please indicate a little about the scoring of the sf-12 - it is unclear if higher or lower scores are better or worse indicators of quality of life.

Line 116 the reference has not formatted through endnote

138 - wonderful that you have “life threatening events after 9/11” as a variable - 

Line 194 - I don’t think this title actually describes what is in the paragraph - it seems that this is not “by exposure” but the group as a whole - ?  It looks like any result there of the exposure - injured vs not - is not here - unless it is not meant to be.

Table 1 - this is very hard to read - too many decimal points - at least for percentages, to the tenth place is easier to read - you also do this rounding in the text of the result, so please do that in the table.  Also, for my eyes, the center justification in the variable column is hard to read - I personally prefer all aligned, perhaps indented below the the name of the variable, and perhaps the name of the variable (like, “Age at 9/11” can be bold?

Discussion section: this section needs to be fleshed out - need more than how the results are consistent or diverge from other studies - a nice discussion of why the study is relevant - synthesis of the results.

Author Response

Response to Reviewer 2 for “Posttraumatic Growth and Quality of Life Among World Trade Center Health Registry Enrollees 16 Years After 9/11”

Comments and Suggestions for Authors

This study is interesting and builds upon a previous study by Registry researchers.  In addition, the theme is very well suited to this special issue.   It is well written by experts who have clearly been working for a long time with the very rich data gathered from members of the WTC Registry.  What follows are some questions to be addressed and points that need to be clarified.  

  1. Line 21: Reference is made to those who were injured on 9/11.  I found out later in the paper that this refers to physical injury, but I think that the word “injury” is used a number of times before this is clarified in the methods section.  This should be clarified, especially since the paper is looking at both physical and mental health quality of life.

Response: We have made this correction throughout the manuscript.

  1. Line 31: “These exposures led to the development of ….”  I am not sure if all of these studies show direct causality - perhaps use the language of “associated with” when the studies do not necessarily show direct causality.

Response: We have made the suggested change.

  1. Line 70: You mention here the reason that this is an important study - this needs to be discussed and elaborated on in the discussion section and noted in conclusion as well.  I am not sure I fully understand how exploring this relationship is important for understanding the long term health impacts and would like this expanded upon later in the paper. 

Response: We added an explanation of the importance of PTG and its association with quality of life measures in the discussion.

  1. Line 75 - One of your goals is to evaluate the association between PTG and mental/physical functioning in a sample of injured and non - injured patients - why would you separate out the two?  I think this needs to be addressed in the introduction as well - the reason that it is believed that these might be different - a theoretical basis would be great, or something that distinguishes these groups in past research?  Lots of research looks at the difference between those caught in the dust cloud or not - why was that NOT chosen?  (Not that it needed to be - but I am using that as an example of why we need to understand the reason that you are looking at these two groups.

Response: We explained the reason for investigating the PTG-SF-12 association separately for injured and non-injured enrollees in the introduction.

  1. Materials and methods: Line 88 - Can you please clarify what the HQoL is?  Is this something that is comprised only of the SF-12, or is the sf-12 embedded in this?  Please describe what this is. And if this particular survey was created by the Registry, should it be cited?

Response: We expanded and clarified on the HQoL survey in the introduction.

  1. Line 110 - Just want to be sure you meant to use reference 20 here rather than 21.

Response: This has been corrected.

  1. OUTCOME - can you please indicate a little about the scoring of the sf-12 - it is unclear if higher or lower scores are better or worse indicators of quality of life.

Response: We expanded on the properties of the SF-12 physical and mental scores in the methods section.

  1. Line 116 the reference has not formatted through endnote

  1. 138 - wonderful that you have “life threatening events after 9/11” as a variable – 

Response: Thank you!

  1. Line 194 - I don’t think this title actually describes what is in the paragraph - it seems that this is not “by exposure” but the group as a whole - ?  It looks like any result there of the exposure - injured vs not - is not here - unless it is not meant to be.

Response: We have corrected this.

  1. Table 1 - this is very hard to read - too many decimal points - at least for percentages, to the tenth place is easier to read - you also do this rounding in the text of the result, so please do that in the table.  Also, for my eyes, the center justification in the variable column is hard to read - I personally prefer all aligned, perhaps indented below the the name of the variable, and perhaps the name of the variable (like, “Age at 9/11” can be bold?

Response: We believe that Table 2 is more appropriate with two decimal points, so we decided to be consistent in Table 1.

  1. Discussion section: this section needs to be fleshed out - need more than how the results are consistent or diverge from other studies - a nice discussion of why the study is relevant - synthesis of the results.

Response: We have added sections to the discussion on how our results fit into the context of research in this field, and on the relevance of our study.

Reviewer 3 Report

The implications of PTG for enrol1lees overall quality of life is important, and there are a couple of suggestions for its improvement:

The literature review is too weak.The authors need to introduce the existing research and methods. And explain the reasons for the methods chosen in this paper.It is recommended that the author add more international literatures to improve this part.

The conclusion part needs to be further improved. In addition to the need to summarize and analyze the results of this study, it is necessary to make a certain outlook on the future research direction.

Author Response

Response to Reviewer 3 for “Posttraumatic Growth and Quality of Life Among World Trade Center Health Registry Enrollees 16 Years After 9/11”

The implications of PTG for enrollee’s overall quality of life is important, and there are a couple of suggestions for its improvement:

Comments:

  1. The literature review is too weak. The authors need to introduce the existing research and methods. And explain the reasons for the methods chosen in this paper. It is recommended that the author add more international literatures to improve this part.

Response: We have added further material to summarize existing research on the association between PTG and Quality of Life.

  1. The conclusion part needs to be further improved. In addition to the need to summarize and analyze the results of this study, it is necessary to make a certain outlook on the future research direction.

Response: We have added further material to summarize our results, the place of our work in the context of existing research, and suggest directions for future research.